# Vertebral Bone Marrow Clot towards the Routine Clinical Scenario in Spine Surgeries: What about the Antimicrobial Properties?

**DOI:** 10.3390/ijms24021744

**Published:** 2023-01-16

**Authors:** Deyanira Contartese, Maria Sartori, Giuseppe Tedesco, Alessandro Gasbarrini, Gianluca Giavaresi, Francesca Salamanna

**Affiliations:** 1Surgical Sciences and Technologies, IRCCS Istituto Ortopedico Rizzoli, 40136 Bologna, Italy; 2Spine Surgery Unit, IRCCS Istituto Ortopedico Rizzoli, 40136 Bologna, Italy

**Keywords:** vertebral bone marrow clot, spinal fusion surgery, mesenchymal stem cells, platelet, antimicrobial properties

## Abstract

Exploring innovative techniques and treatments to improve spinal fusion procedures is a global challenge. Here, we provide a scientific opinion on the ability of a vertebral bone marrow (vBM) clot to provide a local combined delivery system not only of stem cells, signaling biomolecules and anti-inflammatory factors but also of molecules and proteins endowed with antimicrobial properties. This opinion is based on the evaluation of the intrinsic basic properties of the vBM, that contains mesenchymal stem cells (MSCs), and on the coagulation process that led to the conversion of fibrinogen into fibrin fibers that enmesh cells, plasma but above all platelets, to form the clot. We emphasize that vBM clot, being a powerful source of MSCs and platelets, would allow the release of antimicrobial proteins and molecules, mainly cathelicidin LL- 37, hepcidin, kinocidins and cationic host defense peptides, that are per se gifted with direct and/or indirect antimicrobial effects. We additionally highlight that further studies are needed to deepen this knowledge and to propose vBM clot as multifunctional bioscaffold able to target all the main key challenges for spinal fusion surgery.

## 1. Introduction

Spinal fusion (SF) procedures have rapidly increased over the last decade for various debilitating spinal disorders [1,2]. Given the complexities and demands of the procedure, patient selection as well as surgical techniques and instrumentations remain key components to the success of the operation [2]. However, failed SF, or pseudoarthrosis, continues to be a major challenge [1,2]. Therefore, several methods, such as graft materials, cells and growth factors have been intensively investigated to enhance SF. Lately, our research group, for the first time, evaluated the possibility of using a new and advanced formulation of vertebral bone marrow (vBM), the vBM clot [3,4,5]. It consists of a clot naturally formed from bone marrow, which has all the vBM components retained in a matrix molded by the clot. In detail, mesenchymal stem cells (MSCs) from human clotted vBM showed significantly higher growth kinetics in comparison to MSCs from un-clotted vBM as well as greater growth factor expression (e.g., transforming growth factor-β, TGF-β; vascular endothelial growth factor-A, VEGF-A; fibroblast growth factor 2, FGF2), higher osteogenic and chondrogenic differentiation ability, and lower expression of Meis3 and Pbx1 genes, TALE and HOX class genes that negatively regulate the induction, proliferation, differentiation, and maturation of osteoblasts [4]. These results suggest that the cellular source inside the clotted vBM have the best biological properties. Furthermore, the use of clotted vBM not only eliminates the need to concentrate and/or purify vBM but also presents an attractive SF cell therapy strategy able to provide higher “stability” to the graft site in comparison to the existing approaches. Considering these findings and the increasing prevalence of spinal surgery in elderly and super-elderly patients, we assessed the effect of aging on the vBM clot. The study revealed that vBM clot regenerative properties, such as growth factors expression and MSC morphology, viability, surface antigen expression, colony-forming units and osteogenic differentiation ability, were not affected by donor age, as well as Klotho (an aging suppressor gene) and senescence-associated gene expression (interleukin (IL)1-β, IL1-α, IL6, IL8, tumor necrosis factor-α (TNF-α), monocyte chemoattractant protein-1 (MCP)-1, CCL4, CXCL2) [5]. Based on these results and to confirm and strengthen the data on the clinical application of vBM clot for SF procedures, a pilot clinical study on the use of clotted vBM in 10 patients with degenerative spine diseases is ongoing at our institution (Ethical Committee approval n. 587/2020/Sper/IORS). Preliminary clinical data demonstrated that a good bony fusion as well as an improved Oswestry Disability Index (ODI) and Visual Analog Scale (VAS) were present in all patients at an average follow-up of 3 and 6 months [6,7]. In addition to this regenerative capacity proved by the vBM clot, we believe that this autologous bioscaffold may be able to target all the main key challenges for SF surgery. First, the vBM clot acts as an osteogenic and osteoinductive three-dimensional (3D) bioscaffold containing pluripotent mesenchymal and hematopoietic stem cell that work synergistically to foster bone formation and regeneration [5]; second, platelets degranulation in the vBM clot allow the release of many biomolecules (α-granules, platelet-specific proteins, cytokines/chemokines, growth factors, coagulation factors, adhesion molecules) that might promote an early vascularization, vital for bone homeostasis, healing, regeneration and hardware osseointegration [3,5]; third, mesenchymal progenitors in vBM clot modulates inflammation through a paracrine immunomodulatory effect allowing an optimal transient stage of acute inflammation, key element for a successful bone healing [3]; fourth, an additional powerful element of vBM clot could be represented by the critical role of the coagulation cascade and of the bone marrow mesenchymal stem cells (BMSCs) in the early innate immune system activation and in its involvement in stressing and eliminating bacteria [3,4,5]. We assume that the ability of vBM clot to provide a local combined delivery system of stem cells, signaling biomolecules and anti-inflammatory and antibacterial factors enclosed by a matrix molded by the clot represent an advanced and simple strategy to meet the main clinical needs of SF. As the identification of the mechanism of action of each therapeutic approach remains a major challenge for clinical translational and considering that spinal infection is one of the major complications after SF surgery, in this opinion we hypothesized and discussed the not yet rated antimicrobial potential of vBM clot.

## 2. The Antimicrobial Activity of the vBM Clot

The hypothesized mechanisms that underlie the antimicrobial effect of the vBM clot start from the intrinsic basic properties of the BM, which contains MSCs, and from the coagulation process. Additionally, the coagulation factors that led to the conversion of fibrinogen into fibrin fibers that enmesh cells, plasma but above all platelets, to form the clot, were also considered and discussed. Nevertheless, the respective impact and mechanism of action of these components, as well as the possible synergistic effects among them, in the context of antimicrobial activity, are poorly known.

### 2.1. BMSCs Antimicrobial Activity

Despite the demonstrated biologic effect and regenerative properties of MSCs in bone healing and regeneration, few data currently exist on their antimicrobial effect. However, the antimicrobial effect of MSCs has been investigated in other research fields [8,9,10,11,12,13,14,15]. It was shown that MSCs exert powerful antimicrobial effects across indirect and direct mechanisms [8,9,10,11,12,13,14,16,17,18,19]. Indirectly, through their role in (i) the host’s immune response versus pathogens, (ii) active coordination of the pro- and anti-inflammatory elements of the immune system or (iii) increase in phagocyte activity; and directly by peptides and antimicrobial protein (AMP) secretion and by the expression of specific molecules such as indoleamine 2,3-dioxygenase (IDO) and IL17 [8,9,10,11,12,13,14,16,17,18,19]. The AMPs have a discriminatory activity versus a large range of bacteria, yeasts, fungi, viruses and even cancer cells [20]. During infection and inflammation four AMPs are expressed in MSCs, i.e., cathelicidin LL-37, human β-defensin-2 (hBD-2), hepcidin and lipocalin-2 (Lcn2) [9,11,12,16]. The antimicrobial efficacy of AMP-mediated MSCs has been assessed and described for several sources of stromal cells, and BMSCs are the most studied source (Figure 1). 

Two AMPs, i.e., cathelicidin LL-37 and hepcidin, appear to mediate the antimicrobial effect of human BMSCs [9,12,16]. In the bone marrow the LL-37 is expressed in monocytes and neutrophils [22], and in addition to antimicrobial activities it has many other biological activities, such as control of responses to inflammation, as well lipopolysaccharide (LPS)-neutralization [22]. Many of these biological activities produced by LL-37 are mediated by various putative cell membrane channels, intracellular targets, or surface receptors [22]. A purinergic receptor belonging to the ionotropic ATP-gated receptors, the P2X7 receptor (P2X7R), seems to be critical to the bioactivity of LL-37 [22]. P2X7R activation is implicated in the osteogenic differentiation of BMSCs by stimulating the extracellular signal-regulated kinase 1/2 (ERK1/2) and c-Jun NH2-terminal kinase (JNK) signaling pathways in a P2X7R-dependent manner [22]. Concerning the antimicrobial activities of LL-37, they seem to be directed towards several Gram-negative and Gram-positive bacteria, such as *Pseudomonas, Escherichia, Staphylococcus* and *Enterococcus* types [9,12,16]. LL-37 is capable of killing bacteria through direct antibacterial actions, as well as through immunomodulation. Its key mechanism of action is carried out through membrane rupture. The net positive charge of +6 allows LL-37 to bind to the negatively charged bacterial membrane; subsequently, the induction of transmembrane pores lead to a disruption of cell integrity, thus leading to cell lysis and death [9,12,16]. Additional effects of LL-37 are the immunomodulatory, involving both pro-inflammatory and anti-inflammatory responses, which are of key importance to the indirect destruction of bacteria [12]. A supplementary effect on LL-37 is provided by vitamin D3, which appears to be the master regulator of its expression in humans, since the cathelicidin gene-encoding LL-37 has three vitamin D response elements on its promoter [23]. Despite the stimulatory influence of Vitamin D having been extensively studied and analyzed in the field of BMSC osteogenic potential, future studies are mandatory for evaluating if BMSCs’ effects on AMPs expression might also represent a powerful advanced and alternative therapeutic antimicrobial option. With respect to LL-37, in vitro studies demonstrated that human BMSCs and their conditioned medium inhibit the bacterial growth of *E. coli*, *P. aeruginosa*, *S. aureus*, and *S. pneumonia* [9,12]. Furthermore, in an in vivo study set up by using an immunocompetent model of pneumonia by *E. coli*; it was shown that BMSCs treatment after 4 h of *E. coli* produced a sudden reduction in total bacterial counts in lung homogenates and bronchoalveolar lavage fluid compared with the control group [9]. In another in vivo study on a mouse model of cystic fibrosis infected with *P. aeruginosa* and *S. aureus*, treatment with BMSCs resulted in a reduction in colony-forming units for both pathogens in bronchoalveolar lavage fluid. 

Concerning the other AMP that mediates the antimicrobial effect in human BMSCs, i.e., hepcidin, it was reported that it performs, both in vitro and in vivo, a broad spectrum of antimicrobial activities against fungal species and clinically significant bacteria such as *E. coli*, *S. epidermidis*, *S. aureus*, and group B *streptococci* [15,24,25]. Hepcidin is cationic and may adhere to bacterial membranes. Its incorporation into bacterial membranes may cause disintegration of the lipid bilayer and cell rupture [15]. Furthermore, endogenous expression of hepcidin by myeloid cells, i.e., macrophages and neutrophils, was demonstrated by in vitro and in vivo studies, where these cell types produced hepcidin in response to bacterial pathogens in a Toll-like receptor 4 (TLR4)-dependent way [26,27]. Despite there being no studies on hepcidin production in vBM, these myeloid cell types were also present in bone marrow.

However, it is important to underline that the antimicrobial properties of human BMSCs are not limited to cathelicidin LL-37 and hepcidin action. BMSCs—because of a significant increase in the expression of indoleamine 2,3-dioxygenase and after stimulation with inflammatory cytokines—display a cell-autonomous, broad-spectrum antimicrobial effector function directed against *S. aureus*, *S. epidermidis*, *E. faecium*, Group B *streptococci*, protozoal parasites and viruses [8]. These results suggest an antimicrobial activity that BMSCs may perform during bone infection or the potential to enhance antibiotic activity, considering that there are also no limits to the concomitant treatment of BMSCs based on a particular class of antibiotics.

### 2.2. Coagulation Process: Platelets and Coagulation Factors of Antimicrobial Activity

During the coagulation process, a dense cascade of chemical reactions takes place, and among all of them, platelet degranulation is of essential importance [28]. Platelets are small, disc-shaped, anucleated cells, arranged by basic element, i.e., plasma membrane, the open cannular system, a dense tubular system, a spectrin-based membrane skeleton and an actin-based cytoskeleton network [28]. Furthermore, platelets have a peripheral band of microtubules and several organelles such as α granules, dense granules, peroxisomes, lysosomes and mitochondria [28]. In addition to platelets role in coagulation, homeostasis and innate immune response, a wealth of evidence suggests that they also play a key role in countering microorganisms. Microorganisms can interact with platelets using several mechanisms of action and the binding to platelets can either be a direct interaction or an indirect interaction [29,30,31,32,33,34,35] (Figure 2). 

When a bacterial adhesin binds directly to a platelet receptor, a direct interaction occurs, while when a bacterial adhesin binds to a plasma protein that connects the bacteria to a specific receptor on the platelet surface, an indirect interaction arises. The platelet–microorganism interaction evolves through several progressive phases: (1) direct contact, (2) morphogenesis, (3) early aggregation and (4) permanent aggregation [29,30,31,32,33,34,35,37]. During these phases, platelets have a key shape change, i.e., from discoid to amoeboid with several pseudopodia. This transition typically occurs prior to platelet aggregation, leading to the organization of platelet microtubules such that granules are mobilized from the platelet perimeter into the cytoplasm [29,30,31,32,33,34,35]. This organization precedes platelet degranulation and the secretion of a variety of host defense molecules.

Over the course of infection, platelets secrete specific molecules with microbicidal activities that include kinocidins (antimicrobial chemokines), previously called thrombocidins, cationic host defense peptides (CHDPs) and more recently RNase7 [29,30,31,32,33,34,35]. 

Kinocidins display modular functional domains that can be arranged to act autonomously to facilitate cooperative, synergistic host defense functions. They are larger and have greater structural complexity than traditional antimicrobial peptides and, differently to them, that are confined within phagocytes or expressed on the mucosa, kinocidins are processed immediately into the bloodstream, thus showing relatively lower host toxicity [37]. Individual structural domains in kinocidins offer complementary microbicidal and leukocyte-enhancing functions, thus being multifunctional immune effector molecules that manage molecular and cellular host defense against infection [37] (Figure 3).

The first human kinocidins were detected in the α-granules of platelets: thrombocidin-1 (TC-1) and thrombocidin-2 (TC-2). Both kinocidins, derived from platelet basic protein (PBP, CXCL7), are reduced variants in two amino acids (Ala-Asp) in the amino-terminal region of CXC chemokines and carry out powerful antibacterial activity against *B. subtilis*, *E. coli* and *S. aureus*, and *C. neoformans* [37]. Subsequently, other platelet-derived antimicrobial peptides were purified, and were the chemoattractant peptides (CXC and CC chemokines), e.g., platelet factor-4 (PF-4; CXCL4), regulated on activation normal T cell expressed and secreted (RANTES or CCL5), CTAP-3, PBP, thymosin beta 4 (Tβ-4), fibrinopeptide (FP)-A and FP-B. They employed their antimicrobial activity prevalently against *E. coli* and *S. aureus* [39]. In human PF4, in platelet microbicidal proteins (PMP)-1 (ser-PMP-1) and tPMP-1 (asp-PMP-1), the 1) N-terminal anionic region, with a CXC motif; 2) an intermediate domain, which conforms a β-sheet, antiparallel motif; 3) a C-terminal cationic domain, comprising an α-helix motif consistent with peptides showing direct microbicidal activity; and 4) a 3D structure stabilized by two disulfide bridges were identified and studied [40]. These domains have different functions, half the molecule (residues 1–37) has minimal antimicrobial activity, while the other half (residues 38–74) reduces the growth of *S. aureus, S. typimurium* and *C. albicans* [41].

Despite these data, it is important to emphasize that several discrepancies regarding the microbicidal activity of platelet kinocidins exist among studies [42,43]. These differences are potentially due to the experimental conditions or to the microorganism used; to date, however, the antimicrobial activity of human platelets is now well-recognized. Since there are numerous kinocidins in human platelets, with this opinion we summarize their most important microbicidal activity in Table 1.

CHDPs are small antimicrobial peptides endowed with multiple biological functions. They can act on essential intracellular processes (protein, DNA, RNA synthesis) and can cause damage or lead to bacterial membrane permeabilization, entering the cytoplasm of target cells [50,51]. Cathelicidins and defensins are among the main CHDPs present in humans [52,53,54,55]. Several studies have also highlighted the presence of CHDPs in human platelets. In particular, defensins such as human β-defensins (HBD)-2 and -3 are stored within platelet α-granules and released when they recognize platelet activation-inducing components during vascular damage, i.e., thrombin, adenosine diphosphate and collagen, or by the presence of microbial components [55,56,57,58]. Platelet CHDPs showed relevant microbicidal function for multiple bacterial strains. Defensins can destroy bacteria or prevent their growth through the direct membrane disruption and the inhibition of bacterial cell wall synthesis [55,56,57,58]. They can also decrease bacterial infection by neutralizing toxins [55,56,57,58]. In detail, defensins such as HBD-1 present in the cytoplasm of platelets are released by permeabilizing agents present on the bacterial membrane. HBD-1, once released by platelets, inhibits the growth of bacteria, such as *E. coli* and *S. aureus*, by killing them directly or through the induction of neutrophil extracellular traps (NETs). Similarly, as also previously described for BMSCs, cathelicidins have a broad inhibition spectrum, including bacteria, fungi and viruses [59,60]. 

CHDPs in human platelets can also stimulate platelet activation and aggregation. Human neutrophil peptide-1 (HNP-1) leads to an enhancement in fibrinogen binding, enhance surface expression of activated glycoprotein GP IIb/IIIa, thrombospondin 1 (TSP-1), CD62P, CD63 and soluble CD40L secretion, thus inducing platelet aggregation by creating amyloid-like structures, which can bind microorganisms [61,62].

Recent data have shown that platelets in basal conditions or when infected with M. tuberculosis express RNase7. It is a member of the RNase A superfamily with a cationic domain, which exhibits microbicidal activity versus a wide range of pathogens, and with a powerful ribonuclease adept in degrading RNA [62]. At this step, further studies are needed to better understand RNase7 antimycobacterial activity.

Finally, recent evidence shows how the coagulation factors themselves, in particular factors VII, IX and X, possess powerful antibacterial activity, acting as antimicrobial proteins, especially against Gram-negative bacteria. Coagulation factors, after the activation, are cleaved into a light chain (LC) and a heavy chain (HC), each involved in specific actions and functions [63]. A serine protease activity is associated with HC and its involvement in coagulation cascade initiation, while the antibacterial activity is instead related to the light chain [64]. Indeed, Chen et al. demonstrated how lipopolysaccharide, one of the major components of bacterial outer membrane envelope, can be damaged by the LC-triggered hydrolysis, thus inducing great danger to the Gram-negative bacteria’s survival. Studies performed by Chen et al. showed how LC hydrolysis action was effective against several Gram-negative bacteria—also including drug-resistant (XDR) pathogens, both in vitro and in vivo—in protecting BALB/c mice models against XDR *P. aeruginosa* PA4 from severe infection [63].

### 2.3. Clinical Consequences of vBM Clot Antimicrobial Activity in Spine Surgery

Spine infection can be a critical complication after surgery in both the short and long term, with an incidence that ranges between 0.6% and 18% [65,66]. Infection certainly represents a high-impact event for the patient, but also has considerable management that rotates around several debridements and targeted antibiotic therapies [44]. Infection also significantly increases healthcare costs: the literature reports that the direct costs associated to the treatment of an infected patient can vary between USD 4000 for the treatment of an infected wound to USD 38,000 depending on the type and severity of the infection [67,68]. However, in some cases these approaches are not successful and patient management becomes drastically complicated, with devastating effects on their quality of life. In this context, the vBM clot could represent a natural 3D matrix able to deliver not only osteocompetent cells, biomolecules and anti-inflammatory factors, but also antibacterial molecules already present in the clot itself, thus delivering a safe, sustainable, high-quality and patient-friendly treatment with a maximum health care impact. However, the main obstacle to vBM clot use as an innovative and advanced antimicrobial approach in SF procedures is the absence of studies that demonstrate its antibacterial properties. Thus, we are now planning a platform of studies to evaluate the antibacterial effects of vBM clots against the most common Gram-positive and Gram-negative bacteria of spinal infections, i.e., methicillin-sensitive and methicillin-resistant *S. aureus* and *E. coli*. In vitro time–kill, bacterial adhesion assays, production of specific kinocidins (i.e., CCL3, CCL5, CXCL1, CXCL8, CXCL7, CXCL12) and expression of cathelicidin LL-37 and hepcidin will be investigated.

## 3. Conclusions

Finding an optimal autologous graft substitute able to target all the main key challenges of SF surgery is still a challenging task in bone tissue engineering. In the scenario previously described, the vBM clot, a powerful source of MSCs and platelets, leads to the release of antimicrobial proteins and molecules, i.e., cathelicidin LL-37, hepcidin, kinocidins and CHDPs, that are *per se* endowed with direct and/or indirect antimicrobial effects. Obviously, there is still a long way to go before discovering and understanding the real antimicrobial activity due to these proteins and molecules present in the clot and their impact on immune response. However, the combination of the antibacterial features with the powerful properties of vBM may represent a real, new, cost-effective strategy to deeply explore and so counteract the greatest threat represented by bacterial infections and antibiotic resistance.

## Figures and Tables

**Figure 1 ijms-24-01744-f001:**
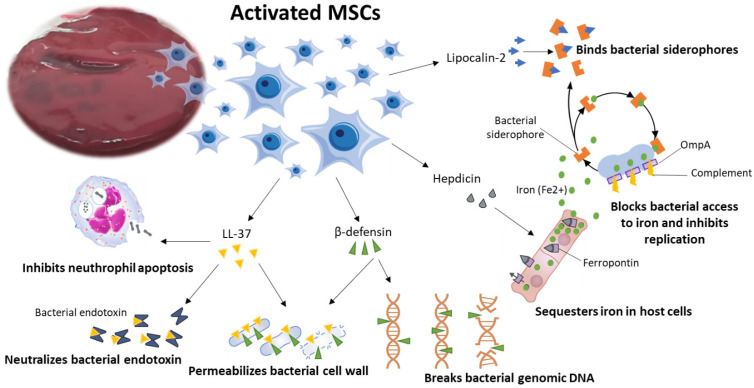
Schematic representation of potential direct antimicrobial activity of BM clot MSCs. Adapted from Shaw et al. [21] by GIMP 2.10.32 software.

**Figure 2 ijms-24-01744-f002:**
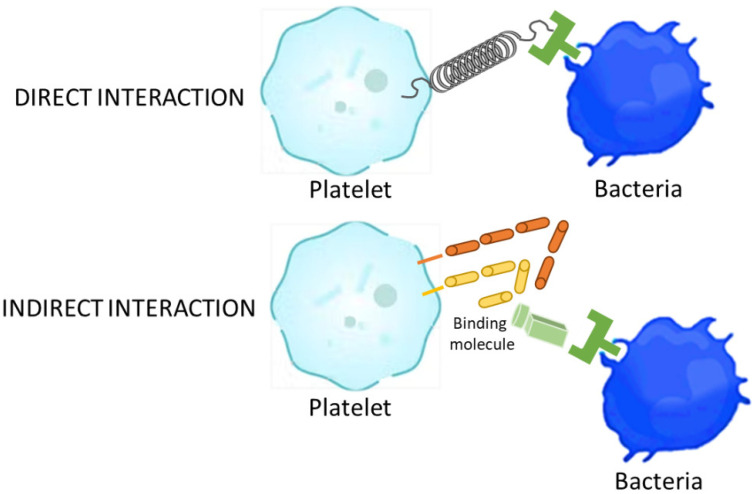
General mechanisms of platelets bacterial interaction (direct and indirect). Adapted from Fogagnolo et al. [36] by GIMP 2.10.32 software.

**Figure 3 ijms-24-01744-f003:**
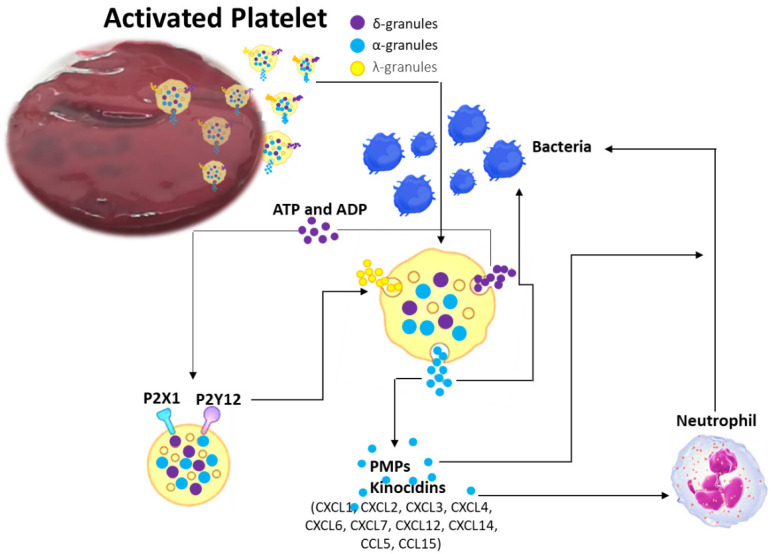
Schematic representation of potential antimicrobial activity of BM clot platelet. Adapted from Yeaman [38] by GIMP 2.10.32 software.

**Table 1 ijms-24-01744-t001:** Key kinocidins in human platelets and target microorganisms. Adapted from Aquino-Domínguez et al. [44].

Kinocidins	Target Microorganisms
CXCL1	*E. coli*, *S. aureus*, *S. typhimurium*, *C. albicans* [43,45]
CXCL2	*E. coli*, *S. aureus* [43]
CXCL3	*E. coli*, *S. aureus* [43]
CXCL4	*E. coli*, *S. aureus*, *S. typhimurium*, *C. albicans* [41,45]
CXCL6	*N. gonorrhoeae*, *E. faecalis*, *P. aeruginosa*, *S. pyogenes*, *S. dysgalactiae subsp*, *S. aureus*, *E. coli*, *B. subtilis* [46,47]
CXCL7	*E. coli*, *S. aureus*, *C. neoformans* [39]
CXCL7 (fragment TC-1)	*E. coli*, *B. subtilis*, *C. neoformans*, *S. aureus* [37,48]
CXCL7 (fragment TC-2)	*E. coli*, *S. aureus*, *B*. *subtilis* [37]
CXCL12	*E. coli*, *S. aureus* [42]
CXCL14	*E. coli*, *S. aureus*, *E.coli*, *C. albicans* [42,49]
CCL5	*E. coli*, *S. aureus*, *S. typhimurium* [39,45]
CCL15	*E. coli*, *S. aureus* [39]
CCL17	*E. coli*, *S. aureus* [39]

## Data Availability

Not applicable.

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
