# Peer review of "Vertebral Bone Marrow Clot towards the Routine Clinical Scenario in Spine Surgeries: What about the Antimicrobial Properties?"

_ijms, 2023, doi:10.3390/ijms24021744_

Round 1
Reviewer 1 Report
Dear Editor,
I read the manuscript titled “Vertebral bone marrow clot towards the routine clinical scenario in spine surgery protocols with great interest. Only to fuse? ”, which has been submitted to the International Journal of Molecular Sciences.
The authors carried out a narrative review on the role of bone marrow clots (BMc) in spinal fusion. The study focused on the antimicrobial properties of the BMc and presented the relevant underlying pathways.
The topic is interesting and relevant to the daily practice of spine surgeons. The manuscript is generally well-written, and the appropriate references, tables, and figures support it.
However, the current draft has some significant methodological limitations.
1. The authors should specify the study design of the current article in the methodology section.
2. Equally important, the authors are advised to describe how they ended up with the gathered evidence. Probably, they performed a literature search. The authors are kindly requested to provide details of the literature search, including search terms, databases, and eligibility criteria, to enhance the transparency and reproducibility of the article.
3. What was the quality of the studies used to support the current draft? I am concerned about the presence of bias in primary studies.
All concerns mentioned above may be resolved after a major revision.
Best regards
Author Response
The authors carried out a narrative review on the role of bone marrow clots (BMc) in spinal fusion. The study focused on the antimicrobial properties of the BMc and presented the relevant underlying pathways.
The topic is interesting and relevant to the daily practice of spine surgeons. The manuscript is generally well-written, and the appropriate references, tables, and figures support it.
However, the current draft has some significant methodological limitations.
- The authors should specify the study design of the current article in the methodology section.
We thank the reviewer very much for the comments and suggestions. As indicated by the reviewer we specified more in detail in the abstract and at the end of the introduction section that the article is an opinion.
- Equally important, the authors are advised to describe how they ended up with the gathered evidence. Probably, they performed a literature search. The authors are kindly requested to provide details of the literature search, including search terms, databases, and eligibility criteria, to enhance the transparency and reproducibility of the article.
As anticipated in the previous answer, our article is a scientific opinion and after the suggestion of the reviewer we emphasized this aspect in the manuscript. Our opinion article presents our view on the strengths and limitations of our starting hypothesis (the ability of vBM clot to provide a local combined delivery system of stem cells, signaling biomolecules and anti-inflammatory and antibacterial factors enclosed by a matrix molded by the clot that represent an advanced and simple strategy to meet the main clinical needs of SF), to clarify to researchers and clinicians where we stand in this field. For this reason, the article does not present a 'literature review' structure, since it intends to help assess where we are heading and what we should do to meet research trends in this specific field. Thus, there is not a bibliographic search strategy that follows the PRISMA criteria as the reviews, but it is based on the knowledge of the authors on this topic and obviously on the most recent scientific literature that the authors constantly evaluate, dealing with this topic from several years.
- What was the quality of the studies used to support the current draft? I am concerned about the presence of bias in primary studies.
We thank the reviewer for the comment. Being this article an opinion, the quality of the studies reported in the manuscript was not assessed. This type of article does not want to give a certainty and/or a final datum but wants to lead the researcher to do new research in this research field, considering aspects not analyzed and/or neglected until now. The field of research analyzed in this opinion is until now unknown and based on scientific evidence related to other therapeutic options, therefore our opinion wants to try to open a new field of research by emphasizing what is present in the vBM clot and how it could act.
Reviewer 2 Report
Summary:
This is an interesting opinion article regarding the clinical utility and underlying mechanistic description of the use of vertebral bone marrow clot to improve clinical outcomes of spinal fusion surgeries. The authors describe the biological properties of mesenchymal stem cells and the antimicrobial properties of platelets. The authors summarized the possible mechanism for the regenerative properties of the bone marrow mesenchymal stem cells and focused their discussion mainly on the potential antimicrobial properties of the vertebral bone marrow clot. Extensive literature review was performed by the authors to dissect the antimicrobial properties of bone marrow mesenchymal stem cells, platelets and coagulation factors. The future aims of this research group to perform further analysis of the antimicrobial properties of the vertebral blood clot was clearly delineated in line 270 – 276. Overall, this is an interesting article and further characterization of the types of cells within a vertebral bone marrow clot and the antimicrobial profile of these different cell types would be of great scientific interest to the medical community.
Strengths:
1) The author’s previous work which was cited provides strong scientific evidence of the biological properties of the bone marrow mesenchymal stem cells.
2) Extensive literature review by the authors presents a strong argument for the antimicrobial properties of bone marrow mesenchymal stem cells.
3) Excellent figures that depict the proposed mechanism of antimicrobial properties of bone marrow mesenchymal cells and platelets (Figures 1-3).
Weakness:
1) Title could be modified to be clearer: Vertebral bone marrow clot towards the routine clinical scenario in spinal surgeries.
2) Line 54- 59: The authors describe an ongoing clinical study with a statement of improvement in the Oswestry Disability Index (ODI) and Visual Analog Scale (VAS) without further elaboration on what improvements were noted. It would be ideal to cite the references for both these scoring systems (ODI -Fairbank JC, Pynsent PB. The Oswestry Disability Index. Spine (Phila Pa 1976). 2000 Nov 15;25(22):2940-52; discussion 2952. doi: 10.1097/00007632-200011150-00017. PMID: 11074683 ; VAS - Knop C, Oeser M, Bastian L, Lange U, Zdichavsky M, Blauth M. Entwicklung und Validierung des VAS-Wirbelsäulenscores [Development and validation of the Visual Analogue Scale (VAS) Spine Score]. Unfallchirurg. 2001 Jun;104(6):488-97. German. doi: 10.1007/s001130170111. PMID: 11460453.)
3) Line 62-74: References should be provided for each point made to substantiate the argument for the underlying mechanism described.
4) Line 83-85: This sentence is rather long and confusing. Consider using 2 shorter sentences and focus on the bone marrow mesenchymal stem cells and platelets.
5) Line 110-133: Description of the human AMP LL-37 from bone marrow mesenchymal stem cells seems rather limited/ restricted. What cells within humans produce LL-37? Does the vertebral bone marrow clot contain epithelial cells and neutrophils which produce LL-37 (Please see reference Nagaoka, I.; Tamura, H.; Reich, J. Therapeutic Potential of Cathelicidin Peptide LL-37, an Antimicrobial Agent, in a Murine Sepsis Model. Int. J. Mol. Sci. 2020, 21, 5973. https://doi.org/10.3390/ijms21175973)
6) Line 133-136: Both references cited (20 and 21) do not describe the in vitro effects of hepcidin from human BMSCs. Lombardi et al is a review of antimicrobial properties of liver expressed AMP (LEAP-1) hepcidin in general. Abergel et al does not mention hepcidin but describes iron chelating propertis of siderocalin in B.anthracis. Reference 13 (Alcayaga-Miranda et al, 2017) would be a better reference to cite here instead. Hepcidin is expressed in myeloid leukocytes (see references: Peyssonnaux C, Zinkernagel AS, Datta V, Lauth X, Johnson RS, Nizet V. TLR4-dependent hepcidin expression by myeloid cells in response to bacterial pathogens. Blood. 2006; 107(9):3727–32. doi: 10.1182/blood-2005-06-2259 PMID: 16391018 25. Ripley DA, Morris RH, Maddocks SE. Dual stimulation with bacterial and viral components increases the expression of hepcidin in human monocytes. FEMS microbiology letters. 2014; 359(2):161–5. doi: 10.1111/1574-6968.12553 PMID: 25145495 26.) Does the vertebral bone marrow clot contain myeloid leukocytes (neutrophils, monocytes/macrophages, etc) that may be contributing to the antimicrobial activity and not just the BMSCs per se?
7) Line 228-229: Did the authors mean “In detail, …. by permeabilizing agents present on bacterial membrane.”?
8) Line 234: HNP-1 (Is that referring to human neutrophil peptide-1)? Reference 58 and the original paper by the same group should be cited too. (Valle-Jiménez, X.; Ramírez-Cosmes, A.; Aquino-Domínguez, A.S.; Sánchez-Peña, F.; Bustos-Arriaga, J.; Romero-Tlalolini, M.D.; Torres-Aguilar, H.; Serafín-López, J.; Aguilar Ruíz, S.R. Human platelets and megakaryocytes express defensin alpha 1. Platelets 2019, 31, 344–354).
9) Acknowledgements for figures if adapted from other sources or the software/website used to draw the figures (e.g Biorender, etc).
10) Acknowledgement if Table 1 was adapted from reference 58.

Author Response
- Title could be modified to be clearer: Vertebral bone marrow clot towards the routine clinical scenario in spinal surgeries.
We modified the title as suggested by the reviewers: “Vertebral bone marrow clot towards the routine clinical scenario in spine surgeries. What about the antimicrobial properties?”
- Line 54- 59: The authors describe an ongoing clinical study with a statement of improvement in the Oswestry Disability Index (ODI) and Visual Analog Scale (VAS) without further elaboration on what improvements were noted. It would be ideal to cite the references for both these scoring systems (ODI -Fairbank JC, Pynsent PB. The Oswestry Disability Index. Spine (Phila Pa 1976). 2000 Nov 15;25(22):2940-52; discussion 2952. doi: 10.1097/00007632-200011150-00017. PMID: 11074683 ; VAS - Knop C, Oeser M, Bastian L, Lange U, Zdichavsky M, Blauth M. Entwicklung und Validierung des VAS-Wirbelsäulenscores [Development and validation of the Visual Analogue Scale (VAS) Spine Score]. Unfallchirurg. 2001 Jun;104(6):488-97. German. doi: 10.1007/s001130170111. PMID: 11460453.)
As suggested by the reviewer we added the references on VAS and ODI (references 6 and 7).
- Line 62-74: References should be provided for each point made to substantiate the argument for the underlying mechanism described.
As suggested, we added references for each point in the sentence from line 63 to line 75.
- Line 83-85: This sentence is rather long and confusing. Consider using 2 shorter sentences and focus on the bone marrow mesenchymal stem cells and platelets.
As suggested, we spitted the sentence in two shorter sentences.
- Line 110-133: Description of the human AMP LL-37 from bone marrow mesenchymal stem cells seems rather limited/ restricted. What cells within humans produce LL-37? Does the vertebral bone marrow clot contain epithelial cells and neutrophils which produce LL-37 (Please see reference Nagaoka, I.; Tamura, H.; Reich, J. Therapeutic Potential of Cathelicidin Peptide LL-37, an Antimicrobial Agent, in a Murine Sepsis Model. Int. J. Mol. Sci. 2020, 21, 5973. https://doi.org/10.3390/ijms21175973)
As recommended by the reviewer we expanded the description of the human AMP LL-37 from bone marrow mesenchymal stem cells and added in the description the reference suggested (Lines 112-121).
- Line 133-136: Both references cited (20 and 21) do not describe the in vitro effects of hepcidin from human BMSCs. Lombardi et al is a review of antimicrobial properties of liver expressed AMP (LEAP-1) hepcidin in general. Abergel et al does not mention hepcidin but describes iron chelating propertis of siderocalin in B.anthracis. Reference 13 (Alcayaga-Miranda et al, 2017) would be a better reference to cite here instead. Hepcidin is expressed in myeloid leukocytes (see references: Peyssonnaux C, Zinkernagel AS, Datta V, Lauth X, Johnson RS, Nizet V. TLR4-dependent hepcidin expression by myeloid cells in response to bacterial pathogens. Blood. 2006; 107(9):3727–32. doi: 10.1182/blood-2005-06-2259 PMID: 16391018 25. Ripley DA, Morris RH, Maddocks SE. Dual stimulation with bacterial and viral components increases the expression of hepcidin in human monocytes. FEMS microbiology letters. 2014; 359(2):161–5. doi: 10.1111/1574-6968.12553 PMID: 25145495 26.) Does the vertebral bone marrow clot contain myeloid leukocytes (neutrophils, monocytes/macrophages, etc) that may be contributing to the antimicrobial activity and not just the BMSCs per se?
As suggested, we replaced references 20 and 21 with reference by reference by Alcayaga-Miranda et al. (now reference 15) and added the references suggested by the reviewer (now references 25-26) with a discussion of these studies since neutrophils and monocytes/macrophages are also present in bone marrow (Lines 150-154).
- Line 228-229: Did the authors mean “In detail, …. by permeabilizing agents present on bacterial membrane.”?
We corrected the sentence as suggested by the reviewer.
- Line 234: HNP-1 (Is that referring to human neutrophil peptide-1)? Reference 58 and the original paper by the same group should be cited too. (Valle-Jiménez, X.; Ramírez-Cosmes, A.; Aquino-Domínguez, A.S.; Sánchez-Peña, F.; Bustos-Arriaga, J.; Romero-Tlalolini, M.D.; Torres-Aguilar, H.; Serafín-López, J.; Aguilar Ruíz, S.R. Human platelets and megakaryocytes express defensin alpha 1. Platelets 2019, 31, 344–354).
As supposed by the reviewer HNP-1 is refer to Human neutrophil peptide-1, we specified it in the manuscript. As suggested by the reviewer we also added the original paper linked to reference.
- Acknowledgements for figures if adapted from other sources or the software/website used to draw the figures (e.g Biorender, etc).
As suggested, we added acknowledgements for figures adapted from other sources and the software used to draw the figures.
- Acknowledgement if Table 1 was adapted from reference 58.
We added acknowledgements for Table 1.
Reviewer 3 Report
Summary of study
This is a review article that discusses the properties of vertebral bone marrow (vBM) clot and the proposed mechanisms for improving outcomes of spinal fusion procedures. The authors extend our existing knowledge of some of the properties of vBM clots (namely the presence of stem cells, signaling molecules and anti-inflammatory factors) to discuss their antimicrobial properties. Certain molecules have been implicated to exert antimicrobial effects, particularly cathelicidin LL-37, hepcidin, kinocidins and cationic host defense proteins.
General concept comments
The manuscript is clear and is presented in a well-structured manner. Its main strength is that it tackles and reviews an unexplored property of vBM clots. Since one of the most devastating complications of spinal fusion surgery is infection, this topic is of high interest for molecular scientists as well as clinicians and surgeons.
Comments
- Please revise to improve punctuation and correct grammar mistakes. There are quite a few scattered throughout the manuscript
- TITLE: Although the title is good, but it does not quite reflect the content of the paper. I would suggest that the title includes something about “antimicrobial properties” of vBM since this is the main theme of the article
- Page 3, line 133: Please include that the antimicrobial effect of hepcidin has been demonstrated in vivo as well (PMID 26293821 and PMID 28655927).
- Page 7, line 246: I realize that you referenced Chen et al later in the paragraph, but please include this reference after the first line where you report that factors VII, IX and X possess a powerful antibacterial activity.
- Page 7, line 260: If this is accurate, please report in that paragraph that there have been no studies to date that examined the antimicrobial effects of vBM in spine surgery.
- Page 7, line 260: Please also include the significant increase in cost of care by $4,067 if an episode of infection develops (PMID 22045005)
Author Response
- Please revise to improve punctuation and correct grammar mistakes. There are quite a few scattered throughout the manuscript
As suggested by the reviewer we checked and corrected the manuscript for grammar mistakes.
- TITLE: Although the title is good, but it does not quite reflect the content of the paper. I would suggest that the title includes something about “antimicrobial properties” of vBM since this is the main theme of the article
We modified the title as suggested by the reviewers: “Vertebral bone marrow clot towards the routine clinical scenario in spine surgeries. What about the antimicrobial properties?”
- Page 3, line 133: Please include that the antimicrobial effect of hepcidin has been demonstrated in vivo as well (PMID 26293821 and PMID 28655927).
As suggested, we include that the antimicrobial effect of hepcidin has been demonstrated in vivo and added the recommended references (references 23-24).
- Page 7, line 246: I realize that you referenced Chen et al later in the paragraph, but please include this reference after the first line where you report that factors VII, IX and X possess a powerful antibacterial activity.
We added the reference number after the sentence.
- Page 7, line 260: If this is accurate, please report in that paragraph that there have been no studies to date that examined the antimicrobial effects of vBM in spine surgery.
As suggested, we reported that there have been no studies that examined the antimicrobial effects of vBM in spine surgery at page 8, lines 293-295.
- Page 7, line 260: Please also include the significant increase in cost of care by $4,067 if an episode of infection develops (PMID 22045005)
As suggested, we include a sentence on the cost of care at page 8 (Lines 291-296) and add the reference indicated by the reviewer (reference 65).